# The Association between Medroxyprogesterone Acetate Exposure and Meningioma

**DOI:** 10.3390/cancers16193362

**Published:** 2024-09-30

**Authors:** Russell L. Griffin

**Affiliations:** Department of Epidemiology, School of Public Health, The University of Alabama at Birmingham, Birmingham, AL 35294-0022, USA; russellg@uab.edu; Tel.: +1-205-975-3037

**Keywords:** medroxyprogesterone acetate, meningioma, contraceptive, hormonal

## Abstract

**Simple Summary:**

Medroxyprogesterone acetate is a synthetic progesterone commonly used as a contraceptive in the United States, used by up to 25% of women aged 18–49 who have had intercourse. Progesterone is reported to be associated with the increased growth of meningiomas, a type of central nervous system tumor that can affect the meninges of the brain or spine, and recently an article utilizing a population of French women noted a significant association between medroxyprogesterone use and cerebral meningioma risk, particularly for patients using it for two years. The current study supported the findings of the prior published research, finding an increased chance of cerebral meningioma for those with use of medroxyprogesterone acetate; this association became stronger with longer duration of use. There were no associations observed for spinal meningiomas.

**Abstract:**

Background/Objectives: Medroxyprogesterone acetate (MPA) is a synthetic progesterone that is most commonly used as a contraceptive. MPA acts by binding to the progesterone receptor of the hypothalamus, and this receptor has been found to be important in the pathophysiology of meningiomas. Recent research has reported an increased association between the use of MPA and intracranial meningioma, though the literature is mostly limited by low numbers of meningioma cases and low exposure to MPA. The objective of the current study is to build upon the previously published literature utilizing a large database from the United States. Methods: Utilizing a large commercial insurance database, the current matched case–control study identified meningioma cases using ICD-10 codes from hospital data and MPA exposure, as established from pharmaceutical claims data. Controls were matched 10:1 to cases based on age, year of enrollment, and duration of enrollment. A conditional logistic regression estimated odds ratios (ORs) for the association between MPA exposure and the odds of developing a meningioma. Results: Among 117,503 meningioma cases and 1,072,907 matched controls, oral MPA exposure was not associated with odds of meningioma; however, injection MPA exposure was associated with a 53% increased odds of being a case (OR 1.53, 95% CI 1.40–1.67). This association was specific to cerebral meningiomas (OR 1.68, 95% CI 1.50–1.87), an association that became stronger with a longer duration of use of injection MPA. Conclusions: The current results are consistent with the prior literature, which reports an association between injection exposures to MPA and a stronger association with increasing use of MPA. Women should be cautioned about the prolonged use of MPA, and future research should examine whether the extended use of MPA is associated with the meningioma grade.

## 1. Introduction

Medroxyprogesterone acetate (MPA) is part of the family of progestins. It is a synthetic progesterone that is used for a variety of indications. In oral form, progestins can be used alone or alongside an estrogen in combined oral contraceptives (COCs) [1]. Other indications for progestins include hormone replacement therapy (HRT), treating menstrual disorders, and, for MPA, treating endometrial carcinomas [2]. MPA is administered daily via oral routes in 2.5, 5, and 10 mg doses, or it can be administered as depot MPA (dMPA) through a 150 mg/mL intramuscular injection or a 104 mg/0.65 mL subcutaneous injection. In the United States, the prevalence of MPA use in injection form is estimated to be 3% of all women [3] and 25% of women aged 15–49 who have had intercourse [4]. MPA binds to the progesterone receptor of the hypothalamus, which, in turn, prevents the secretion of gonadotropin-releasing hormone, resulting in a cascade of events leading to the inhibition of follicular maturation, the prevention of ovulation, and the thickening of the cervical mucus, which, in turn, prevents sperm mobility.

Progesterone is thought to be an important factor in the pathophysiology of meningiomas. Meningiomas are mostly slow-growing, benign tumors [5] that account for approximately one-third of all intracranial tumors and are known to be more prevalent among females, with the ratio approaching 3:1 female–male following puberty [6,7]. Further, it is believed that sex hormones play an important part in meningioma growth, as an increase in the size of meningiomas has been reported during pregnancy, believed to be potentially due to an increase in hormones such as progesterone [8]. Further, the literature has reported a high prevalence (38–88%) of progesterone receptors in meningiomas [9,10]. The higher prevalence of progesterone receptors has been reported to be greatest among medial skull-base meningiomas, compared to lateral skull-base, non-skull-base, and spinal meningiomas [11]. They are also more prevalent among females [12,13,14] and in individuals younger than 50 years [15].

Recently, Roland et al. [16] examined the association between a wide array of progestogens and the risk of developing a meningioma, especially related to the use of the progestogens cyproterone acetate, chlormadinone acetate, and nomegestrol acetate [17,18]. A novel finding of over 5.5-fold increased odds of meningioma for use of MPA was reported; however, the prevalence of the use of MPA in France, where the study was set, is low, and only 20 individuals reported any MPA exposure.

The objective of the current study is to utilize a large national database of insurance records to examine whether the previously reported association with MPA and meningioma is observed in a different population of women with a higher prevalence of MPA use.

## 2. Materials and Methods

### 2.1. Data Sources

Data for the current study were derived from the IBM MarketScan database for the years 2006–2022. The MarketScan database is one of the largest private-insurance claims databases in the United States. The data include medical and pharmaceutical data derived from a variety of sources (notably hospital discharges, electronic medical records, and claims data, which were used for the current study), and they are de-identified prior to being made available to end-users. The current study was exempted from Institutional Review Board review due to the de-identified nature of the data.

### 2.2. Study Design and Variables

For this matched case–control study, data were collected on insurance enrollment (i.e., date of enrollment start and end), demographics (i.e., sex and age), clinical diagnoses (via ICD-9 and ICD-10 codes), and pharmaceutical claims. Cases were defined as females aged 18 or older who were diagnosed with either cerebral meningioma or spinal meningioma based on the presence of an ICD-9 or ICD-10 CM diagnosis code for either malignant or benign meningioma (Table 1). For each case, up to 10 female controls were matched (with replacement) based on age ± 1 year and exact year of enrollment. Controls had to be enrolled for at least as long as the case on the encounter date on which the first ICD code for meningioma was documented (Figure 1).

For all cases and matched controls, an Elixhauser comorbidity score was calculated by first collecting all documented ICD diagnosis codes up to three years prior to the case date. ICD-9 CM codes were converted to ICD-10 CM codes, and these codes were used to determine whether the study subject had a prior medical history of the Elixhauser comorbidities. An unweighted Elixhauser comorbidity score was calculated as the sum of the individual’s comorbidities, and the score was categorized as 0, 1, 2, 3, or ≥4 comorbidities. For the purposes of the current analysis, age was categorized in approximately 20-year age groups: 18–39 years, 40–59 years, and ≥60 years. The primary exposure was having a dispensed prescription for MPA. A prescription duration variable was created by summing the total time between the first dispensed MPA prescription and the date of the last documented dispensing (prior to the case date). Duration was categorized a priori as no exposure, <1 year, 1–2 years, 2–3 years, and >3 years, based on the recommendation to not use dMPA for more than two years unless no other birth control option is suitable (2), and to make the current duration associations comparable to those reported by Roland et al. [16]. Separate variables were created for oral exposure (via tablet, both alone and combined with an estrogen) and injection (including both the 150 mg intramuscular dose and the 104 mg subcutaneous dose).

### 2.3. Statistical Analysis

Using a complete-case analysis, conditional logistic regression was used to estimate odds ratios (ORs) and associated 95% confidence intervals (CIs) to determine the association between a prescription drug claim for MPA and the odds of a meningioma diagnosis. Adjusted models were created by including age (to account for residual confounding after matching) and the Elixhauser comorbidity score. Additional analyses examined the duration of the MPA prescription (based on prescription claims data), to examine a potential dose–response relationship; for these models, ECI category was entered as both a categorical variable (to estimate ORs by category) and, separately, as a continuous variable (to test for a linear trend across categories). Separate models were created for cerebral and spinal meningiomas. In a post hoc analysis, logistic regression models were created for cases of unspecified meningioma site and their matched controls. SAS v9.4 was used for all analyses, and *p*-values < 0.05 were considered statistically significant.

## 3. Results

### 3.1. Bivariate Analysis

A total of 117,503 cases and 1,072,907 matched controls were included in the analysis; of the matched controls, 97.5% (n = 1,257,989) were matched only once, 2.4% (31,298) were matched twice, and 0.1% were matched three (n = 947), four (n = 41), or five (n = 1) times. There was no meaningful difference in age between cases and controls (mean 58.2 vs. 58.1 years, respectively) (Table 2). Cases had a higher average unweighted Elixhauser comorbidity score (mean 1.70 vs. 0.97, *p* < 0.0001) than controls. Notable comorbidities for which cases had a higher prevalence included deficiency anemias (9.2% vs. 5.5%, *p* < 0.0001), malignant solid tumor without metastasis (9.2% vs. 2.7%, *p* < 0.0001), cerebrovascular disease (8.2% vs. 2.3%, *p* < 0.0001), and other neurologic disorders (15.1% vs. 1.3%, *p* < 0.0001).

### 3.2. Overall Associations

For all meningiomas, the prevalence of oral exposure to MPA was similar between cases (2.38%) and controls (2.29%) (Table 3). In both crude and adjusted models, MPA exposure was not associated with being a case (adjusted OR 0.97, 95% CI 0.93–1.01); this null association persisted across all duration categories (Table 3). The prevalence of injection exposure to MPA was nearly twice as high among cases (0.67%) than controls (0.39%); MPA exposure was associated with 76% increased odds of being a case (OR 1.76, 95% CI 1.63–1.90), an association that persisted in the adjusted model (OR 1.53, 95% CI 1.40–1.67). There was evidence of dose–response by exposure duration, with an increasing association with increasing duration (linear trend, *p* < 0.0001). In particular, the association in adjusted models ranged from 23% increased odds of being a case (OR 1.23, 95% CI 1.10–1.38) for ≤1 year exposure duration, to 2.5-fold increased odds for exposure durations >3 years (OR 2.50, 95% CI 2.06–3.04). 

### 3.3. Associations by Meningioma Type

By site of meningioma, there was no association between spinal meningioma and both oral MPA exposure (adjusted OR 1.01, 95% CI 0.85–1.19) and injection MPA exposure (OR 0.77, 95% CI 0.53–1.12) (Table 4). For cerebral meningioma, oral MPA exposure was not associated with the odds of being a case (OR 0.99, 95% CI 0.94–1.04); however, injection MPA exposure was associated with 68% increased odds (OR 1.68, 95% CI 1.50–1.87). Further, there was a significant linear trend by duration (*p* < 0.0001), with the association ranging from 35% increased odds of being a case for exposure durations ≤1 year (OR 1.35, 95% CI 1.17–1.56), to over three-fold increased odds for exposures greater than 3 years (OR 3.24, 95% CI 2.49–4.21).

### 3.4. Post Hoc Analysis for Unspecified Meningioma Type

In post hoc analyses, the association between dMPA exposure and unspecified meningioma was similar to the reported associations for meningioma sites combined (Table 5). This is not unexpected, given that the unspecified sites are likely a similar mix of cerebral and spinal meningiomas, as observed for cases with a reported meningioma site. 

## 4. Discussion

In this matched case–control study, MPA was associated with increased odds of being a meningioma case with evidence of a dose–response relationship with increasing duration of use. This association was notably specific to injection exposure to MPA and cerebral meningiomas. No association was observed for oral MPA exposure or for spinal meningiomas (for both oral and injection MPA exposure).

Though the current literature on the association between MPA and meningioma is limited, the current results are consistent with the published literature. In a 2023 case series of skull-base meningiomas among women in Pennsylvania, the authors reported a mean duration of dMPA use of 15.5 years, with a range of 6–26 years [19]. This supports the current study’s observation of the strongest association occurring among women who had prolonged dMPA exposure. In the same case series, the authors noted that tumor shrinkage occurred among half of the patients who were instructed to cease their use of dMPA. A 2018 Indonesian case–control study of 101 meningiomas reported over 30-fold increased odds of meningioma among those exposed to dMPA compared to the oral contraceptive (OR 31.49, 95% CI 9.39–105.58). The study also reported over 18-fold increased odds of meningioma for contraceptive use over 10 years (relative to no use); however, this association was not limited to dMPA exposure. Additionally, it should be noted that the associations reported were not adjusted for possible confounding variables, particularly age, which was significantly different between cases and controls. In a slightly larger case–control study utilizing data from the INTERPHONE Study Group from Sweden [20], though injection contraceptive exposure was combined with a subdermal implant and hormonal intrauterine device contraception exposure, a duration of use of at least five years was associated with over two-fold increased odds of meningioma but was not associated with glioma. 

These prior three studies, however, were based on a small number of meningioma cases. In a recent, much larger study (n = 18,061 cerebral meningioma cases), examining exposures to progestogens and odds of cerebral meningioma among women living in France [16], while no association for progestogens overall was detected,5.6-fold increased odds were observed specifically for prolonged MPA use, defined as a dispensation within both years prior to the case index date. The smaller association in the current study relative to the Roland study may be explained by that study’s use of national health system data; its ability to adjust for area of residence; and the small number of subjects exposed to MPA. 

While the current study did not aim to identify a biologic mechanism for the association between MPA and intracranial meningioma, such a mechanism exists. For nearly five decades, it has been known that meningiomas exhibit sex hormone receptors [21,22], and it is believed that these receptors may play a role in the growth of meningiomas [23]. While research on the role of MPA on meningioma risk is limited, research has reported on an increased risk of meningioma associated with exposure to high-dose cyproterone acetate, another progestin used for birth control among other indications [18,24,25]. There is evidence that meningiomas which have developed during exposure to cyproterone acetate exhibit different progesterone receptor patterns than other meningiomas. Further, these meningiomas have been observed to more often be progesterone receptor-positive [26] and more often found on the skull base [27], a location that has been found to more often have sites of PI3KCA mutations [28]. Though research on the biological mechanism of MPA and meningioma is limited to date, Pletzer et al. [29] reported on progesterone receptor affinity of 11 progestins; of these, MPA had the fourth highest affinity. It is, therefore, plausible that meningioma growth could be stimulated by the presence of MPA, increasing the likelihood of a meningioma diagnosis, much like with cyproterone acetate.

### Limitations

The current study should be viewed in light of certain strengths and limitations. First, the study is strengthened by the use of a large, commercial insurance-based database that allowed for a larger number of MPA-exposed study subjects than prior studies, as well as a larger number of meningioma cases compared to prior studies. Second, the utilization of a nested case–control design minimizes the potential for exposure recall bias, as the temporality of exposure relative to meningioma diagnosis can be determined through pharmaceutical claims and hospital discharge data. That said, there is potential for exposure misclassification as the case date—to which the MPA exposure is determined—is the date of clinical diagnosis and may not be representative of the true date of meningioma formation. Secondly, our associations were adjusted for age and comorbidity status, but residual confounding may still be present and could explain part of the observed increased associations. As a limitation inherent to MarketScan data, the study subjects were mostly individuals at large companies with private insurance; therefore, the study’s results may not be generalizable to those who have other types of private insurance, public insurance, or are uninsured. There is, however, no reason to suspect a differential association between MPA exposure and risk of meningioma by insurance status. This study additionally utilized ICD codes to identify meningioma cases; these codes do not provide detail regarding the grade and pathology of the diagnosed meningiomas. Future research should expand upon this limitation by examining whether MPA is associated with an increased risk of certain cerebral meningioma pathologies. Finally, the current study was only able to assess whether MPA exposure increased the risk of meningioma and could not assess other important clinical characteristics such as tumor size or WHO grade.

## 5. Conclusions

The current study reports an increased association between dMPA and meningioma, with the association being limited to cerebral meningioma. Though meningiomas are often benign and have an over 90% five-year survival rate [30], the first line of treatment is often surgery, and the meningiomas can decrease the quality of life through impaired neurologic function and potential for malignant behavior [31], particularly following surgery [32]. Future research should assess whether MPA is associated with larger meningiomas or higher-grade meningiomas.

## Figures and Tables

**Figure 1 cancers-16-03362-f001:**
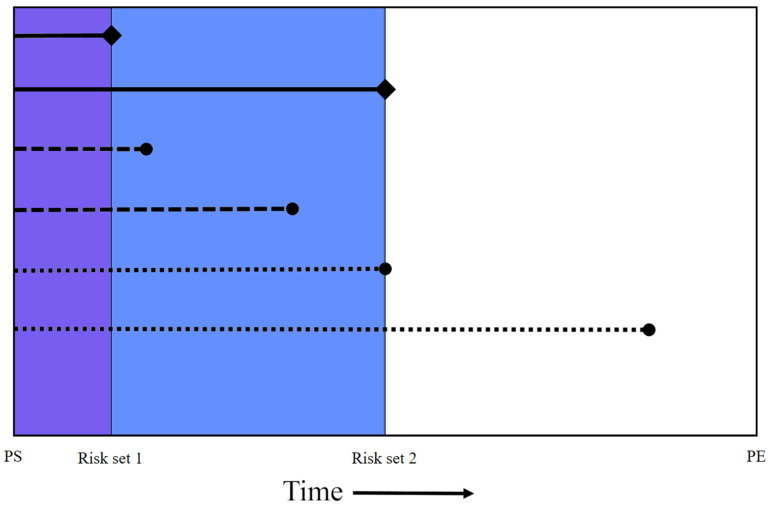
Nested case–control design. At the time, a case (diamond end caps) occurred, matched controls (circle end caps) were selected from all persons under follow-up (i.e., from insurance policy start [PS] date to policy end [PE] date). Only exposures prior to the case occurrence were counted for matched controls (i.e., purple for risk set 1 [dashed lines] and both purple and blue areas for risk set 2 [dotted lines]).

**Table 1 cancers-16-03362-t001:** ICD-9 CM and ICD-10 CM codes included for cerebral and spinal meningioma.

	Benign	Malignant	Unknown Behavior
Cerebral	225.2, D32.0	192.1, C70.0	D42.0
Spinal	225.4, D32.1	192.3, C70.1	D42.1
Unspecified site	D32.9	C70.9	237.6, C70, D32, D42, D42.9

**Table 2 cancers-16-03362-t002:** Comparison of age and comorbidities of meningioma cases and matched controls.

	Cases (n = 117,503)	Controls (n = 1,072,907)	*p*-Value *
Age			
Mean years (SD)	58.2 (15.8)	58.1 (15.0)	<0.0001
Categorical (%)			
18–39	12,262 (10.2)	130,948 (10.9)	0.0132
40–59	54,299 (45.2)	559,512 (46.6)	
60+	53,671 (44.6)	510,223 (42.5)	
Elixhauser comorbidities			
Mean unweighted score (SD)	1.70 (2.95)	0.97 (2.07)	<0.0001
AIDS	120 (0.1)	794 (0.1)	0.0011
Alcohol abuse	1047 (0.9)	6436 (0.6)	<0.0001
Deficiency anemias	10,798 (9.2)	58,624 (5.5)	<0.0001
Autoimmune conditions	3497 (3.0)	21,920 (2.0)	<0.0001
Chronic blood loss anemia	1307 (1.1)	6981 (0.7)	<0.0001
Leukemia	481 (0.4)	1170 (0.1)	<0.0001
Lymphoma	919 (0.8)	2652 (0.2)	<0.0001
Metastatic cancer	3262 (2.8)	4806 (0.4)	<0.0001
Solid tumor without metastasis, in situ	1311 (1.2)	8625 (0.8)	<0.0001
Solid tumor without metastasis, malignant	10,868 (9.2)	28,764 (2.7)	<0.0001
Cerebrovascular disease	9.619 (8.2)	24,333 (2.3)	<0.0001
Heart failure	3543 (3.0)	20,703 (1.9)	<0.0001
Coagulopathy	2912 (2.5)	10,024 (0.9)	<0.0001
Dementia	3597 (3.1)	15,259 (1.4)	<0.0001
Depression	12,787 (10.9)	72,519 (6.8)	<0.0001
Diabetes with chronic complications	6840 (5.8)	46,503 (4.3)	<0.0001
Diabetes without chronic complications	8573 (7.3)	60,952 (5.7)	<0.0001
Drug abuse	1695 (1.4)	10,102 (0.9)	<0.0001
Complicated hypertension	4563 (3.9)	27,324 (2.5)	<0.0001
Uncomplicated hypertension	22,124 (18.8)	162,503 (15.1)	<0.0001
Mild liver disease	5309 (4.5)	25,650 (2.4)	<0.0001
Moderate/Severe liver disease	267 (0.2)	1383 (0.1)	<0.0001
Chronic pulmonary disease	9620 (8.2)	64,689 (6.0)	<0.0001
Neurological disorders affecting movement	2385 (2.0)	10,120 (0.9)	<0.0001
Other neurologic disorders	17,685 (15.1)	14,047 (1.3)	<0.0001
Seizures and epilepsy	6790 (5.8)	8.384 (0.8)	<0.0001
Obesity	14,003 (11.9)	98,468 (9.2)	<0.0001
Paralysis	2729 (2.3)	4040 (0.4)	<0.0001
Peripheral vascular disorders	6905 (5.9)	38.609 (3.6)	<0.0001
Psychoses	4295 (3.7)	24,362 (2.3)	<0.0001
Pulmonary circulation disorders	1371 (1.2)	7341 (0.7)	<0.0001
Moderate renal failure	3330 (2.8)	22,122 (2.1)	<0.0001
Severe renal failure	803 (0.7)	5212 (0.5)	<0.0001
Hypothyroidism	10,751 (9.1)	77,834 (7.3)	<0.0001
Other thyroid disease	5824 (5.0)	30,220 (2.8)	<0.0001
Peptic ulcer with bleeding	927 (0.8)	5012 (0.5)	<0.0001
Valvular disease	6252 (5.3)	32,503 (3.0)	<0.0001
Weight loss	3510 (3.0)	14,993 (1.4)	<0.0001

* Estimated from conditional logistic regression.

**Table 3 cancers-16-03362-t003:** Odds ratios * (ORs) and associated 95% confidence intervals (CIs) for the association between medroxyprogesterone exposure and meningioma.

	Cases (n = 117,503)	Controls (n = 1,072,907)	Crude OR (95% CI)	Adjusted † OR (95% CI)
Oral Exposure				
Any (%)	2892 (2.38)	27,526 (2.29)	1.05 (1.01–1.09)	0.97 (0.93–1.01)
Duration (%)				
No exposure	118,730 (97.62)	1,174,447 (97.71)	Referent	Referent
≤1 year	2082 (1.71)	19.078 (1.59)	1.09 (1.04–1.14)	1.00 (0.95–1.05)
>1–2 years	293 (0.24)	3056 (0.25)	0.96 (0.85–1.08)	0.89 (0.78–1.01)
>2–3 years	210 (0.17)	2122 (0.18)	0.99 (0.86–1.14)	0.92 (0.79–1.07)
>3 years	307 (0.25)	3270 (0.27)	0.94 (0.84–1.06)	0.89 (0.78–1.01)
Injection Exposure				
Any (%)	813 (0.67)	4652 (0.39)	1.76 (1.63–1.90)	1.53 (1.40–1.67)
Duration (%)				
No exposure	120,809 (99.33)	1,197,321 (99.61)	Referent	Referent
≤1 year	457 (0.38)	3112 (0.26)	1.47 (1.33–1.62)	1.23 (1.10–1.38)
>1–2 years	106 (0.09)	590 (0.05)	1.84 (1.50–2.27)	1.74 (1.38–2.18)
>2–3 years	80 (0.07)	334 (0.03)	2.39 (1.87–3.06)	2.30 (1.74–3.02)
>3 years	170 (0.14)	616 (0.05)	2.84 (2.39–3.37)	2.50 (2.06–3.04)

* Estimated from conditional logistic regression. † Adjusted for age and unweighted Elixhauser comorbidity score.

**Table 4 cancers-16-03362-t004:** Odds ratios *† (ORs) and associated 95% confidence intervals (CIs) for the association between medroxyprogesterone exposure and meningioma by site of meningioma.

	Cerebral Meningioma			Spinal Meningioma		
	Cases (n = 87,455)	Controls (n = 866,759)	OR (95% CI)	Cases (n = 6871)	Controls (n = 65,880)	OR (95% CI)
**Oral Exposure**						
Any (%)	1983 (2.27)	18,939 (2.19)	0.99 (0.94–1.04)	183 (2.66)	1659 (2.52)	1.01 (0.85–1.19)
Duration (%)						
No exposure	85,472 (97.73)	847,820 (97.82)	Referent	6688 (97.34)	64,221 (97.48)	Referent
≤1 year	1427 (1.63)	13,135 (1.52)	1.02 (0.96–1.08)	126 (1.83)	1150 (1.75)	1.02 (0.83–1.24)
>1–2 years	204 (0.23)	2163 (0.25)	0.90 (0.77–1.04)	21 (0.31)	185 (0.28)	0.90 (0.55–1.47)
>2–3 years	155 (0.18)	1525 (0.18)	0.96 (0.80–1.14)	14 (0.20)	138 (0.21)	0.98 (0.54–1.76)
>3 years	197 (0.23)	2116 (0.24)	0.91 (0.78–1.06)	22 (0.32)	186 (0.28)	1.11 (0.68–1.79)
**Injection Exposure**						
Any (%)	480 (0.55)	2626 (0.30)	1.68 (1.50–1.87)	38 (0.55)	353 (0.54)	0.77 (0.53–1.12)
Duration (%)						
No exposure	86,975 (99.45)	864,133 (99.70)	Referent	6833 (99.45)	65,527 (99.46)	Referent
≤1 year	273 (0.31)	1796 (0.21)	1.35 (1.17–1.56)	27 (0.39)	245 (0.37)	0.81 (0.52–1.25)
>1–2 years	60 (0.07)	346 (0.04)	1.68 (1.24–2.27)	3 (0.04)	38 (0.06)	0.59 (0.17–1.99)
>2–3 years	47 (0.05)	189 (0.02)	2.39 (1.68–3.41)	1 (0.01)	32 (0.05)	0.29 (0.04–2.21)
>3 years	100 (0.11)	295 (0.03)	3.24 (2.49–4.21)	7 (0.10)	38 (0.06)	1.04 (0.41–2.65)

* Estimated from conditional logistic regression. † Adjusted for age and unweighted Elixhauser comorbidity score.

**Table 5 cancers-16-03362-t005:** Odds ratios * (ORs) and associated 95% confidence intervals (CIs) for the association between medroxyprogesterone exposure and meningioma for cases of unspecified meningioma site and their matched controls.

	Cases (n = 27,296)	Controls (n = 269,334)	Crude OR (95% CI)	Adjusted † OR (95% CI)
**Oral Exposure**				
Any (%)	726 (2.66)	6928 (2.57)	1.04 (0.97–1.13)	0.91 (0.84–0.99)
Duration (%)				
No exposure	26,570 (97.3)	262,406 (97.43)	Referent	Referent
≤1 year	529 (1.94)	4793 (1.78)	1.10 (1.00–1.20)	0.96 (0.87–1.06)
>1–2 years	68 (0.25)	708 (0.26)	0.96 (0.75–1.23)	0.85 (0.65–1.12)
>2–3 years	41 (0.15)	459 (0.17)	0.89 (0.65–1.23)	0.76 (0.54–1.08)
>3 years	88 (0.32)	968 (0.36)	0.91 (0.73–1.13)	0.81 (0.64–1.03)
**Injection Exposure**				
Any (%)	295 (1.08)	1673 (0.62)	1.76 (1.55–2.00)	1.49 (1.29–1.72)
Duration (%)				
No exposure	27,001 (98.92)	267,661 (99.38)	Referent	Referent
≤1 year	157 (0.58)	1071 (0.40)	1.45 (1.22–1.73)	1.14 (0.94–1.38)
>1–2 years	43 (0.16)	206 (0.08)	2.14 (1.54–2.98)	2.15 (1.48–3.11)
>2–3 years	32 (0.12)	113 (0.04)	2.73 (1.82–4.08)	2.86 (1.79–4.57)
>3 years	63 (0.23)	283 (0.11)	2.26 (1.71–2.98)	2.00 (1.47–2.73)

* Estimated from conditional logistic regression. † Adjusted for age and an unweighted Elixhauser comorbidity score.

## Data Availability

Data are not available for sharing from the author and must instead be obtained from the parent company that owns the MarketScan research database.

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
