# Peer review of "The Association between Medroxyprogesterone Acetate Exposure and Meningioma"

_cancers, 2024, doi:10.3390/cancers16193362_

Round 1

Reviewer 1 Report

Comments and Suggestions for Authors

The manuscript entitled "The association between medroxyprogesterone acetate exposure and meningioma" provides interesting findings that MPA exposure is strongly associated with  meningioma grade. The authors also demonstrates that women should be cautioned about prolonged use of MPA. The study is well designed and properly performed. However, there are several points to be addressed.

1. I realized that there is lack of analysis regarding MPA dose and meningioma. Is there any possibility to add that?

2. A brief illustrative figure would be helpful for understand the study design.

3. Since there are some basic research on sex hormone and meningioma, it would be necessary to discuss the underline mechanisms in the discussion part.

4. Is there any relationship between MPA and meningioma grade? Please add specific data.

5. What is the potential relationship between MPA treatment and pathological type of meningioma? Please maybe add this. 

6. The prevalence of various type of meningioma is different. The author should definitely pay attention on the pathological type of meneingioma. 

7. Please add subtitle of the results part for better illustration and improve the readability. 

Author Response

I have attached my comments.

Reviewer 2 Report

Comments and Suggestions for Authors

The authors present an interesting article on a case-control study to try to investigate the role of MPA in the development of nervous system tumors such as meningiomas.

Using a large commercial insurance database may entail a selection bias, since only patients with a certain economic condition that allows them to access that insurance will be included in the study. A comment on this bias should be added in the Discussion.

Despite being a study in which patients are de-identified, it is essential that the study be approved by the Institutional Review Board to verify that the ethical principles of medical research are followed.

The quality of the Tables provided is adequate, as well as the statistical analysis.

The bibliographic references provided are current and relevant to the justification of the study.

Author Response

I have attached my comments.
